# Exploring the Versatility of Microemulsions in Cutaneous Drug Delivery: Opportunities and Challenges

**DOI:** 10.3390/nano13101688

**Published:** 2023-05-21

**Authors:** Zouhair Ait-Touchente, Nadia Zine, Nicole Jaffrezic-Renault, Abdelhamid Errachid, Noureddine Lebaz, Hatem Fessi, Abdelhamid Elaissari

**Affiliations:** 1Univ Lyon, Université Claude Bernard Lyon-1, CNRS, ISA-UMR 5280, 69100 Villeurbanne, France; nadia.zine@univ-lyon1.fr (N.Z.); nicole.jaffrezic@univ-lyon1.fr (N.J.-R.); abdelhamid.errachid@univ-lyon1.fr (A.E.); abdelhamid.elaissari@univ-lyon1.fr (A.E.); 2Univ Lyon, Université Claude Bernard Lyon-1, CNRS, LAGEPP UMR 5007, 69100 Villeurbanne, France; noureddine.lebaz@univ-lyon1.fr (N.L.); hatem.fessi@univ-lyon1.fr (H.F.)

**Keywords:** microemulsions systems, drug delivery, cutaneous drug delivery, microemulsion bioavailability, skin penetration

## Abstract

Microemulsions are novel drug delivery systems that have garnered significant attention in the pharmaceutical research field. These systems possess several desirable characteristics, such as transparency and thermodynamic stability, which make them suitable for delivering both hydrophilic and hydrophobic drugs. In this comprehensive review, we aim to explore different aspects related to the formulation, characterization, and applications of microemulsions, with a particular emphasis on their potential for cutaneous drug delivery. Microemulsions have shown great promise in overcoming bioavailability concerns and enabling sustained drug delivery. Thus, it is crucial to have a thorough understanding of their formulation and characterization in order to optimize their effectiveness and safety. This review will delve into the different types of microemulsions, their composition, and the factors that affect their stability. Furthermore, the potential of microemulsions as drug delivery systems for skin applications will be discussed. Overall, this review will provide valuable insights into the advantages of microemulsions as drug delivery systems and their potential for improving cutaneous drug delivery.

## 1. Introduction

Current skin drug delivery methods play a pivotal role in achieving effective therapeutic outcomes and addressing various dermatological conditions [1,2,3]. Topical administration is a widely utilized approach for delivering drugs to the skin, offering several advantages such as localized treatment, reduced systemic side effects, and enhanced patient compliance [4,5,6]. Traditional methods, such as creams, ointments, and gels, have been employed for decades to deliver drugs to the skin [7]. However, these conventional formulations often face limitations in terms of drug penetration and controlled release [8]. In recent years, advanced skin drug delivery techniques have emerged [9]. One such promising technology is microemulsions (MEs), which have gained significant attention in the pharmaceutical research field due to their unique properties and potential applications in delivering both hydrophilic and hydrophobic drugs [10,11,12,13]. Skin drug delivery is particularly important due to the accessibility and large surface area of the skin, enabling the effective treatment of dermatological conditions, wound healing, and localized therapy [14,15]. Moreover, by delivering drugs directly to the site of action, skin drug delivery can bypass first-pass metabolism and systemic degradation, resulting in enhanced bioavailability and therapeutic efficacy [16]. Exploring innovative approaches for skin drug delivery is crucial in expanding treatment options, improving patient outcomes, and advancing the field of dermatology [2].

Microemulsions are optically transparent [17], thermodynamically stable [10,18], and possess a low interfacial tension, making them an attractive option for drug delivery [19]. They are typically composed of an oil phase, a surfactant, a co-surfactant, and an aqueous phase [17,20]. The small droplet size, typically ranging from 10 to 100 nm, of microemulsions enables efficient drug solubilization and enhanced bioavailability [21,22]. However, the stratum corneum, the outermost layer of the skin, poses a significant barrier for drug penetration [23]. Microemulsions have emerged as a promising approach for enhancing drug delivery through the skin [11,17,21]. Their unique properties play a crucial role in overcoming the skin barrier and facilitating efficient drug absorption [21,24]. The use of microemulsions in cutaneous drug delivery offers several benefits, such as enhanced skin permeation, improved drug stability, and controlled release kinetics [10,24]. By exploiting the optimized formulation and characteristics of microemulsions, researchers can achieve the effective delivery of both hydrophilic and hydrophobic drugs to the target site on the skin [25].

In this comprehensive review, we aim to provide a detailed overview of microemulsions as an advanced drug delivery system, with a particular emphasis on their potential for cutaneous drug delivery. We will discuss the formulation and characterization of microemulsions, including the different types of microemulsions, their composition, stability, and characterization techniques. Moreover, we will highlight the advantages and limitations of microemulsions as a drug delivery system, as well as their potential use in topical formulations. Overall, this review aims to provide valuable insights into the role of microemulsions in overcoming drug delivery challenges and improving cutaneous drug delivery.

## 2. Background and Definition of Microemulsions

Microemulsions are a type of nanocarrier that consist of an oil phase, a surfactant, a co-surfactant, and an aqueous phase [26]. These components are carefully selected and mixed to form a thermodynamically stable system that is optically transparent and has a low interfacial tension [17,19,27]. In addition, microemulsions have several advantages over conventional drug delivery systems [28]. They can improve drug stability, enhance drug permeability, and provide sustained drug release [10,29]. Moreover, microemulsions are versatile and can be used to deliver drugs through various routes, including oral, parenteral, and topical [11].

Microemulsions have been in use for a long time, even before they were officially described. For example, Australian women used a transparent emulsion containing eucalyptus oil, water, soap flakes, and gasoline to wash wool. A patent was filed in the mid-1930s for the creation of a single-phase system by mixing water and oil with the assistance of a surfactant [17]. However, it was not until 1943 that the first academic research on the topic was conducted [30]. In 1959, Schulman et al. coined the term ‘microemulsion’ to describe the spontaneous emulsification process facilitated by a powerful surface-active agent. [31]. These systems typically contain a hydrophobic or a hydrophilic compound, a surfactant, and a cosurfactant [31]. However, the term “microemulsion” has been incorrectly used to describe other systems, such as micelles, reverse micelles, mesophase or liquid crystalline systems, and even coarse emulsions that have undergone micronization, causing confusion [32]. To address this, Danielsson and Lindmann proposed a definition in 1981 that states, *“A microemulsion is a thermodynamically stable isotropic liquid system comprising an aqueous phase, an oil phase, and an amphiphilic substance”* [33]. Microemulsions differ from emulsions in that they are transparent, thermodynamically stable, and contain droplets with sizes smaller than 200 nm, while emulsions are opaque, thermodynamically unstable, and have droplet sizes larger than 200 nm [34,35,36]. Additionally, microemulsions form spontaneously, while energy input is required for emulsions [37]. It is important to note that miniemulsions are a type of emulsion, and should not be confused with microemulsions [38,39].

Microemulsions can be classified based on their structure, which can be water-in-oil (W/O), bicontinuous, or oil-in-water (O/W) (Figure 1) [40]. In a W/O microemulsion, water droplets are dispersed within an oil phase, whereas in an O/W microemulsion, oil droplets are dispersed within a continuous aqueous phase [36]. Bicontinuous microemulsions have a complex, interconnected structure in which both the oil and water phases are continuous and dispersed throughout the system [40]. W/O microemulsions are typically used for delivering hydrophilic drugs [41], while O/W microemulsions are preferred for hydrophobic drugs [42]. Bicontinuous microemulsions have been shown to be effective for delivering both hydrophilic and hydrophobic drugs [43].

## 3. Formulation of Microemulsions

Formulation is a crucial aspect of developing microemulsions for drug delivery applications [44,45]. A well-designed formulation can ensure that the microemulsion system possesses the desired physicochemical properties such as stability, size, and drug loading capacity [46,47]. Typically, microemulsions are composed of four basic components: the oil phase, surfactant, co-surfactant, and aqueous phase [17,48]. The choice of these components depends on the physicochemical properties of the drug and the desired route of administration [49,50].

The oil phase, which can be hydrophobic or hydrophilic, is a key component in determining the stability of the microemulsion [35]. The selection of an appropriate oil phase is based on factors such as solubility of the drug and its compatibility with the surfactant [51]. The surfactant is another critical component, which lowers the interfacial tension between the oil and water phases and stabilizes the microemulsion system [51,52]. The selection of a surfactant depends on its ability to form a monolayer at the oil–water interface and its compatibility with the co-surfactant [53].

The co-surfactant is typically a short-chain alcohol or a glycol, which enhances the solubilization capacity of the surfactant and helps to stabilize the microemulsion [20,54]. The aqueous phase can be composed of water or an aqueous buffer, depending on the desired pH of the system [47,55]. Additionally, the presence of co-solvents, such as ethanol, can improve the solubility of the drug in the microemulsion [56,57].

Various methods can be used for formulating microemulsions, including phase inversion temperature (PIT) [58], spontaneous emulsification [38], and ultrasound-assisted emulsification [59]. In the PIT method, the microemulsion is formed by heating the mixture of oil, surfactant, co-surfactant, and water above the phase inversion temperature, followed by cooling to room temperature [60]. The spontaneous emulsification method involves the gradual addition of the aqueous phase to the mixture of oil, surfactant, and co-surfactant with constant stirring [61]. Ultrasound-assisted emulsification involves the application of high-frequency ultrasound waves to the mixture of oil, surfactant, co-surfactant, and aqueous phase to generate microemulsions [62].

The formulation of microemulsions should consider the desired route of administration, the physicochemical properties of the drug, and the stability of the microemulsion system [63]. The use of suitable oils, surfactants, and co-surfactants can enhance the stability and drug loading capacity of the microemulsion [64]. The selection of an appropriate method for formulation is also crucial in determining the quality of the microemulsion system [65].

### 3.1. Surfactants

When formulating microemulsions for drug delivery, the choice of surfactant is crucial for achieving successful outcomes [49,66]. Several factors should be considered when selecting a surfactant, including microemulsifying properties, compatibility with the route of administration, and the solubility of active ingredients. There are different classes of surfactants (Figure 2) [67]. Ionic and nonionic surfactants are the two main types [68], with ionic surfactants being further divided into anionic, cationic, and amphoteric surfactants based on the dissociation of their hydrophilic group in water [69,70]. Cationic surfactants such as hexadecyltrimethylammonium bromide [71] and dodecyl trimethyl ammonium bromide [72], anionic surfactants such as dioctyl sodium sulfosuccinate [73] and sodium dodecyl sulfate [74], and amphoteric surfactants such as lecithins and phospholipids [75] have been commonly used in studies. Nonionic surfactants, which include a wide range of options such as polysorbate 80 [76], PEG-8 [77], Pluronic F-68 [78] and vitamin E TPGS [79], do not dissociate into ions in aqueous solutions, and are classified based on their specific hydrophilic group [80]. Alkyl polyglycosides such as Oramix CG-110 have gained attention for their excellent biodegradability and renewable sources, making them an attractive option for microemulsion formulation for various applications, including skin delivery [81]. The selection of surfactant should be based on the specific requirements of the intended application [67].

### 3.2. Co-Surfactants

In order to induce a much lower interfacial tension decrease, the addition of a co-surfactant is often necessary in microemulsion formulations [82]. Short-chain alcohols such as ethanol and isopropanol, alkanediols such as propylene glycol, sucrose ethanol blends, medium chain monoglycerides and diglycerides, alkyl monoglycosides, and geraniol are commonly used co-surfactants [83]. Additionally, glycol ethers such as Transcutol^®^ have been shown to have amphiphilic properties and absorption-enhancing capabilities, making them widely used co-surfactants for oral and dermal applications due to their high solubilizing potency [84,85]. Figure 3 provides examples of the most commonly used co-surfactants in microemulsion formulations. The addition of co-surfactants can lead to the thinning of the interfacial film, which prevents the formation of liquid crystalline structures. These co-surfactants are also distributed between the aqueous and oil phases, helping to moderate the hydrophilicity and lipophilicity of the two phases [5,6,13,14,15]. Recent studies have also explored the use of natural co-surfactants such as lecithin and phospholipids for microemulsion formulation, which have been shown to have good biocompatibility and potential for drug delivery applications.

### 3.3. Oil Phase

The oil phase in microemulsions can consist of various types of compounds, including fatty acids such as oleic acid, alcohols such as octanol and decanol, and esters of fatty acids or alcohols such as isopropyl myristate, isopropyl palmitate, ethyl oleate, isostearyl isostearate, and cetearyl octanoate [66]. Medium-chain triglycerides of caprylic or capric acid and triesters of glycerol and acetic acid such as triacetin are also common in oil phase formulations [25,66]. In addition, some terpenes such as limonene, cineole, camphor, and menthol have also been used in recent studies [86]. The selection of the oil phase depends on various factors such as solubility of the drug, targeted application, and stability of the microemulsion [21].

### 3.4. Aqueous Phase

The choice of aqueous phase in microemulsion formulation is crucial for achieving the desired properties and functionality [86]. In addition to water, viscosifiers such as Carbopol^®^ and xanthan gel can be used to control the viscosity and stability of the microemulsion [87]. For microemulsions intended for parenteral or ocular routes, NaCl is added to ensure the osmolarity with plasma and tear fluid [52,88]. Buffer solutions are also added when lecithin is used as a surfactant to maintain pH values between 7 and 8, which is important to protect phospholipids and triglycerides from degradation [89,90]. Absorption enhancers can be added to improve drug delivery [91], but their compatibility with the other components of the microemulsion should be carefully evaluated to avoid chemical incompatibilities that may compromise the stability of the system [92]. Overall, the choice of aqueous phase and any additives used should be carefully considered to achieve optimal stability and functionality of the microemulsion [66].

### 3.5. Active Pharmaceutical Ingredients (API)

The selection of the appropriate active pharmaceutical ingredient (API) is crucial in the formulation of microemulsions for drug delivery applications [17]. The physicochemical properties of the API such as logP, pKa, structure, molecular weight, and presence of ionizable groups can significantly influence the phase behavior and microemulsion structure [93]. For example, ionic surfactants such as dioctyl sodium sulphosuccinate could be affected by the presence of diclofenac sodium hydrochloride, leading to changes in the properties of the resulting microemulsions [93]. On the other hand, certain active ingredients with surface activity can expand the area in which microemulsions are formed [94], while others, such as tricyclic amines, can act as co-surfactants and reduce the amount of surfactant required [49,95].

It is important to note that some lipophilic compounds may also act as oils and compete or be added to the oil phase during microemulsion formation, which may require an adjustment in the amount of surfactant or co-surfactant used [49,95]. Therefore, the selection and study of the API should be considered an essential step in achieving a stable and effective drug delivery system using microemulsions [17,93].

It is also important to consider the issue of using high concentrations of surfactants in microemulsion formulation, as this can lead to potential toxicity concerns [96]. Recent studies have investigated the use of natural surfactants and co-surfactants, as well as the incorporation of stimuli-responsive polymers and nanoparticles, to reduce the surfactant concentration and enhance the stability and performance of microemulsions as drug delivery systems [97,98]. These approaches offer promising directions for future research in microemulsion drug delivery systems [99].

## 4. Characterization of Microemulsions

The importance of microemulsions in the pharmaceutical industry has made their characterization a widely studied aspect in the literature [83]. There are several methods available for the characterization of microemulsions, which can provide information about their droplet size, zeta potential, viscosity, and thermodynamic stability [100,101].

One commonly used method for microemulsion characterization is dynamic light scattering (DLS), which measures the size distribution of the droplets in the microemulsion [102]. DLS is a non-invasive, fast, and reliable method that provides information on the particle size distribution, polydispersity index, and zeta potential of the droplets [103]. Other methods such as transmission electron microscopy (TEM) [104] and cryogenic transmission electron microscopy (Cryo-TEM) can also be used to visualize the microemulsion droplets and their structure [105].

The thermodynamic stability of microemulsions can be assessed using different techniques, such as conductivity measurements, phase behavior studies, and centrifugation tests [54,106,107]. Conductivity measurements can provide information about the amount and type of surfactants present in the microemulsion [54]. Phase behavior studies involve the construction of ternary phase diagrams to identify the regions of microemulsion formation and to determine the optimal composition for the microemulsion [106]. Centrifugation tests can provide information about the stability of the microemulsion by measuring the sedimentation rate of the droplets [107].

Several factors can affect the characterization of microemulsions, including the composition of the microemulsion, the method of preparation, and the storage conditions [93,108]. The method of preparation, such as the order of addition and mixing speed, can also impact the droplet size and stability of the microemulsion [109]. Storage conditions, such as temperature and time, can affect the physical and chemical properties of the microemulsion and alter its stability [37].

The toxicity or biocompatibility of a drug delivery system is a crucial consideration in its development [110]. To assess the suitability of microemulsions as drug carriers, various characterization methods are employed to evaluate their potential toxicity and biocompatibility [26]. Cytotoxicity assays play a significant role in assessing the effects of microemulsions on cell viability and proliferation [111,112]. Techniques such as the MTT assay [113], Alamar Blue assay [114], and LDH release assay [115] are commonly used to determine cell viability and measure the cytotoxic effects of microemulsions. Hemocompatibility assays are employed to evaluate the compatibility of microemulsions with blood components, including assessments of hemolysis, coagulation, platelet activation, and complement activation [116]. Furthermore, skin irritation and sensitization tests, such as the Draize test and patch testing, are conducted to determine the potential of microemulsions to cause skin irritation or allergic reactions [117,118]. Additionally, in vitro and in vivo biocompatibility studies provide insights into the response of living tissues or organisms to microemulsions, including histological analysis, the assessment of inflammatory responses, immunotoxicity evaluations, and systemic toxicity studies in animal models [26,116]. Genotoxicity and mutagenicity assessments are performed to investigate whether microemulsions induce DNA damage or mutations [119,120]. Techniques such as the Ames test [121], micronucleus assay [122], and comet assay [123] are employed for this purpose. Furthermore, the biodegradation and biocompatibility of microemulsions are examined through enzymatic degradation assays and studies on their fate and clearance in biological systems [124,125]. These characterization methods enable researchers to evaluate the toxicity and biocompatibility profiles of microemulsions, providing valuable insights into their safety and potential as effective drug delivery systems.

In conclusion, the characterization of microemulsions is crucial to ensure their efficacy and safety as a drug delivery system. Various methods are available to characterize the microemulsion droplets, and the stability of the microemulsion can be assessed using different techniques. Several factors can affect the characterization of microemulsions, and a thorough understanding of these factors is necessary for the optimal formulation and characterization of microemulsions. It is worth mentioning that there exist a range of techniques that can be utilized for the evaluation of microemulsions’ toxicity and biocompatibility.

## 5. Ternary Phase Diagrams of Microemulsions

A phase diagram is a graphical representation that displays the thermodynamic equilibrium between phases of a system at different compositions, temperatures, and pressures [126]. In the case of microemulsions, a phase diagram is used to determine the region of thermodynamic stability for different microemulsion formulations [126].

The phase diagram for microemulsions typically displays the composition of the oil, surfactant, co-surfactant, and water, as well as the regions of the different types of microemulsions formed [127]. These regions include the oil-in-water (O/W) region, the water-in-oil (W/O) region, and the bicontinuous region, which represents a mixture of both O/W and W/O microemulsions (Figure 4).

The phase behavior of microemulsions is influenced by several factors, including the chemical nature and concentration of the components, temperature, pressure, and the presence of electrolytes [45]. The phase diagram can be used to optimize the formulation of microemulsions, as it enables the selection of the most stable composition and the identification of the boundaries of the different microemulsion types [128].

Various techniques can be used to construct a phase diagram, including the titration method, conductivity measurement, visual observation, and turbidity measurement [129]. These techniques are used to determine the point at which the microemulsion forms and the region of stability [129].

In conclusion, the phase diagram is a crucial tool for the formulation of microemulsions, as it enables the identification of the composition and stability of the microemulsion system. It is important to consider the factors that influence the phase behavior of microemulsions when constructing a phase diagram, and to use appropriate techniques to accurately determine the phase boundaries.

## 6. Life Science Applications of Microemulsions

Microemulsions have been extensively studied in the pharmaceutical research field due to their unique properties and potential applications in drug delivery [19,75]. As mentioned before, their optically transparent and thermodynamically stable nature, coupled with their ability to deliver both hydrophilic and hydrophobic drugs, make them an attractive option for drug delivery.

Microemulsions have shown great promise in overcoming bioavailability concerns and enabling sustained drug delivery (Figure 5). In drug delivery, microemulsions have been used for various applications, including oral, topical, and parenteral administration [11]. Oral microemulsions have been used to enhance the bioavailability of poorly soluble drugs [57]. Topical microemulsions have shown potential for enhancing the delivery of drugs across the skin [21]. Parenteral microemulsions have been used for the delivery of lipophilic drugs and targeting the lymphatic system [130].

Of particular interest in this review is the potential of microemulsions for cutaneous drug delivery [21,41]. Microemulsions have been shown to improve the skin permeation and retention of drugs following the mechanism shown (and explained in the figure caption) in Figure 6 [24]. Additionally, they have been used for the topical delivery of various drugs, including anti-inflammatory active molecules [131], anti-cancer drugs [132], and anti-fungal agents [133]. The ability of microemulsions to deliver drugs through the skin has been attributed to their small droplet size (generally below 100 nm) and the ability to penetrate through the lipid bilayers of the skin [134].

Table 1 presents the compositions and applications of recently developed microemulsions, with a specific focus on their applications in skin delivery. It is evident from the table that the majority of the active pharmaceutical ingredients (APIs) were hydrophobic in nature, and the microemulsions were designed for various applications. The transdermal delivery of drugs via microemulsions offers a convenient method for achieving systemic pharmacological action. In the reviewed microemulsions, water was the most commonly used continuous phase, while isopropylmyristate and isopropylpalmitate were frequently employed as oil phases. In addition to their use as drug delivery systems, microemulsions have also found applications in various other fields [136]. For instance, they have been used as templates for the synthesis of polymer-based nanoparticles and nanocapsules. They have also been used as lubricants, cutting fluids, and in the food industry, including as a nanoreservoir of flavors.

However, it is important to note that the application of microemulsions in drug delivery is still in its early stages, and several challenges remain to be addressed [75]. For instance, the optimization of microemulsion formulations for specific drugs and routes of administration requires a thorough understanding of the factors that affect their stability, such as surfactant and co-surfactant concentrations, and the pH and ionic strength of the system [75]. Furthermore, the toxicity and safety of microemulsions must be thoroughly investigated before their clinical use can be considered [50].

In conclusion, microemulsions are promising drug delivery systems that have shown potential for various applications in pharmaceuticals, including cutaneous drug delivery. Their unique properties and advantages have made them an attractive option for drug delivery, and their potential applications in other fields further highlight their versatility. However, further research is needed to optimize their formulation and characterization, as well as to investigate their safety and toxicity before their clinical use can be fully realized.

## 7. Advantages of Microemulsions as a Drug Delivery System

As previously mentioned, microemulsions have been extensively studied as a drug delivery system due to their unique properties and potential applications in delivering both hydrophilic and hydrophobic drugs. One of the main advantages of microemulsions is their ability to improve drug solubilization and bioavailability, which are major challenges for drug delivery [163]. Their small droplet size enables efficient drug solubilization and enhanced bioavailability, making them an attractive option for drug delivery [163]. Additionally, they have a low interfacial tension and a large interfacial area, which promote drug release and absorption [164]. Compared to other drug delivery systems, microemulsions have several advantages, including their high drug solubilization capacity, low toxicity, and cost-effectiveness [165].

Microemulsions have a high solubilizing power for a variety of molecules, including hydrophilic, lipophilic, and amphiphilic compounds [51,166]. The solubilizing power of microemulsions has been demonstrated for a variety of hydrophobic drugs, such as ibuprofen, cyclosporine A, and curcumin [167]. In addition to their solubilizing power, microemulsions have also been shown to improve the skin permeation of drugs [24]. The small droplet size of a microemulsion allows for enhanced penetration of drugs through the skin barrier [168]. Additionally, the use of surfactants in microemulsions can disrupt the stratum corneum, further enhancing skin permeation [169].

The potential of microemulsions for cutaneous drug delivery is noteworthy, as they have shown efficient skin penetration and drug delivery [21,41]. Microemulsions can penetrate the skin more efficiently than other delivery systems, such as creams and ointments, due to their small droplet size [170]. This makes them a potential candidate for delivering drugs to treat skin diseases and disorders [140]. Overall, the ability of microemulsions to solubilize various molecules and facilitate skin permeation makes them promising delivery systems for a wide range of drugs, including those with poor solubility or low skin permeability.

## 8. Challenges and Future Directions

Despite the numerous advantages of microemulsions as drug delivery systems, there are still several challenges and limitations that need to be addressed [52,165]. One major challenge is the complexity of microemulsion formulation, which requires a thorough understanding of the physicochemical properties of the components and their interactions [171]. Furthermore, the optimization of microemulsion formulations can be time-consuming and costly [172].

Another limitation is the potential toxicity of the surfactants and co-surfactants used in microemulsion formulations [50,96]. Although the toxicity of these components can be minimized by selecting biocompatible surfactants and co-surfactants, there is still a risk of adverse effects, particularly with chronic use [97,98].

In addition, the stability of microemulsions can be affected by environmental factors such as temperature, pH, and ionic strength, which can lead to phase separation or droplet aggregation [35]. Therefore, there is a need for the development of more stable microemulsion formulations that can withstand these challenges.

To overcome these challenges, several potential solutions have been proposed. One strategy is to use natural surfactants and co-surfactants, such as phospholipids and bile salts, which are biocompatible and have lower toxicity compared to synthetic surfactants [97,98]. Another approach is to use nanoparticles, such as liposomes or solid lipid nanoparticles, as carriers for the microemulsion, which can improve stability and enhance drug targeting [173].

Future directions for research in microemulsion drug delivery systems include the development of new methods for characterizing and optimizing microemulsion formulations, as well as the exploration of new applications in different therapeutic areas. Additionally, there is a need for further investigation of the pharmacokinetics and toxicity of microemulsion formulations, particularly with chronic use.

In conclusion, microemulsions represent promising drug delivery systems with numerous advantages, including high solubilization capacity, improved bioavailability, and versatility in delivering both hydrophilic and hydrophobic drugs. However, there are still challenges and limitations that need to be addressed, such as formulation complexity, potential toxicity, and stability issues. With the development of new strategies and the advancement of research in this field, microemulsions have the potential to become widely used drug delivery systems for various therapeutic applications.

To illustrate the advantages and limitations of microemulsions for drug delivery, Table 2 is provided in this review. This table highlights the potential benefits of microemulsions, such as improved drug permeability, sustained drug release, and versatile drug delivery routes. It also summarizes the limitations of microemulsions, including potential toxicity, formulation complexity, and scaling-up challenges. The table serves as a useful reference for researchers and practitioners working in the field of drug delivery. By carefully considering the benefits and limitations of microemulsions, researchers can develop more efficient and effective drug delivery systems that can improve patient outcomes.

## 9. Conclusions

In this comprehensive review, we have discussed the potential of microemulsions as innovative cutaneous delivery systems. We began by providing a brief overview of drug delivery systems in pharmaceuticals and explaining the characteristics of microemulsions, including their composition and properties. We then discussed the formulation of microemulsions and the various techniques used for their preparation.

We also highlighted the importance of characterizing microemulsions and discussed the different methods for their characterization, as well as the factors affecting their characterization. Furthermore, we provided an overview of the potential applications of microemulsions in drug delivery, with a particular emphasis on their use in cutaneous drug delivery.

The advantages of microemulsions as drug delivery systems were also discussed, including their high stability, efficient drug solubilization, and potential for targeted drug delivery. We also compared microemulsions with other drug delivery systems, such as liposomes and nanoparticles.

Despite their many benefits, microemulsions face some challenges and limitations in drug delivery. We reviewed these challenges, including the complexity of formulation and potential toxicity concerns. We also discussed potential solutions to overcome these challenges, including the use of safer surfactants and co-surfactants.

In conclusion, microemulsions represent a promising and innovative drug delivery system for cutaneous applications. Their unique properties, such as high stability and efficient drug solubilization, make them an attractive option for targeted drug delivery. However, further research is needed to address the challenges and limitations of microemulsions and to explore their full potential in drug delivery. With ongoing advancements in microemulsion technology, it is clear that these systems have a bright future in the field of pharmaceutical research and development.

## Figures and Tables

**Figure 1 nanomaterials-13-01688-f001:**
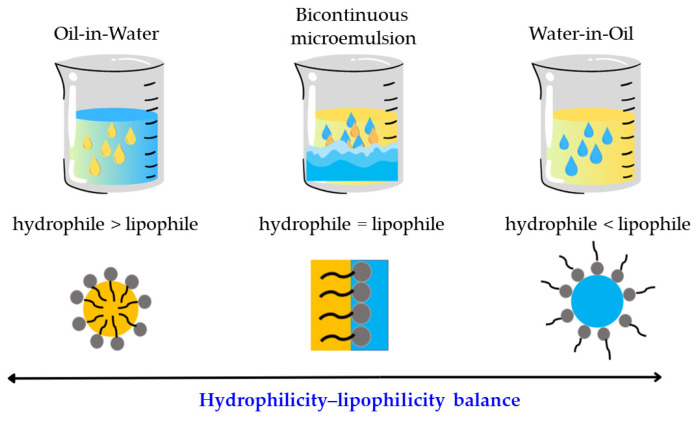
Different configurations of MEs: water-in-oil, bicontinuous, and oil-in-water.

**Figure 2 nanomaterials-13-01688-f002:**
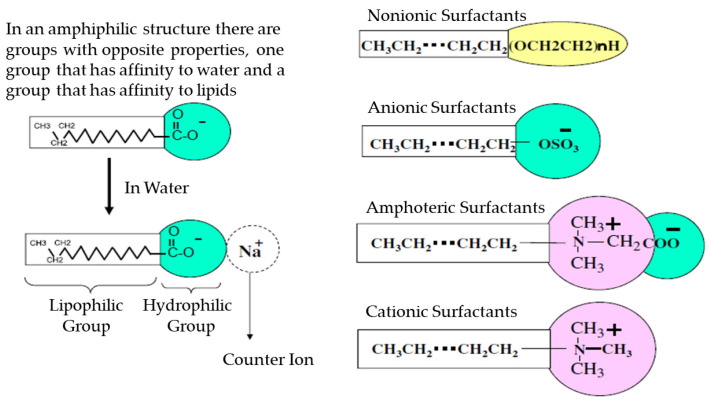
An overview of surfactants: Their structures and classifications. Reproduced with permission [67]. Copyright 2023, Elsevier.

**Figure 3 nanomaterials-13-01688-f003:**
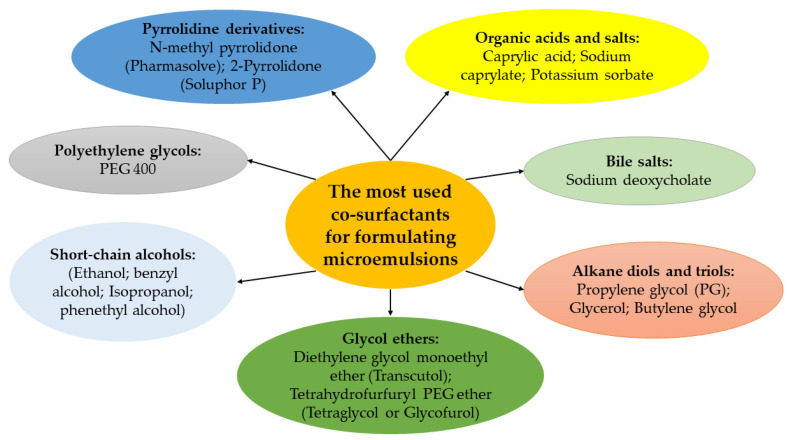
Commonly used co-surfactants with examples employed for formulating microemulsions.

**Figure 4 nanomaterials-13-01688-f004:**
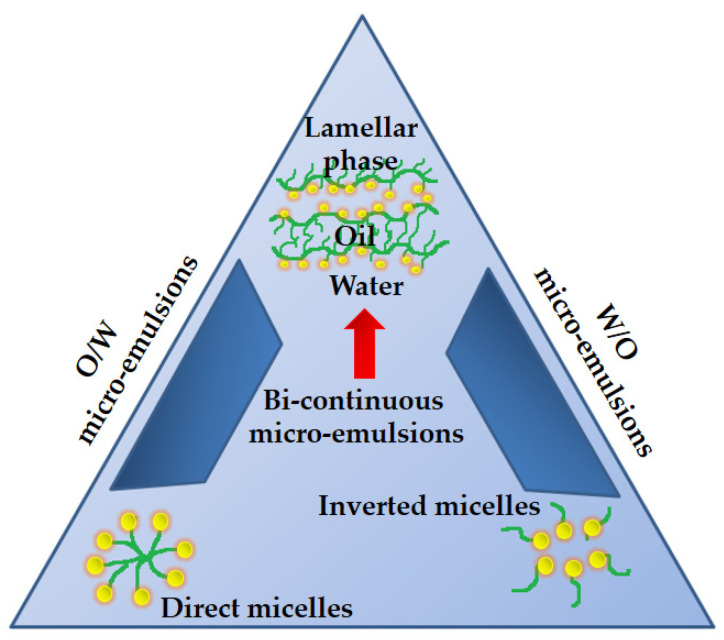
Illustration of the phase diagram indicating the microemulsion formation regions.

**Figure 5 nanomaterials-13-01688-f005:**
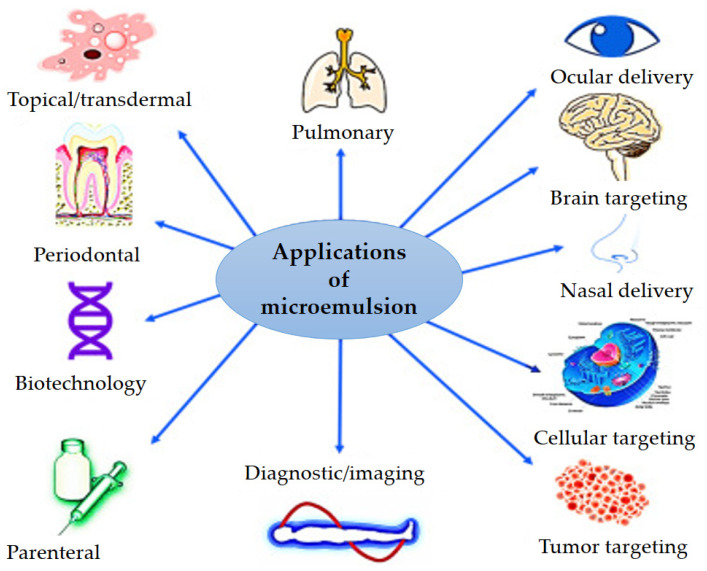
Versatile applications of microemulsions in drug and gene delivery. Reproduced with permission from [11]. Copyright 2023, Elsevier.

**Figure 6 nanomaterials-13-01688-f006:**
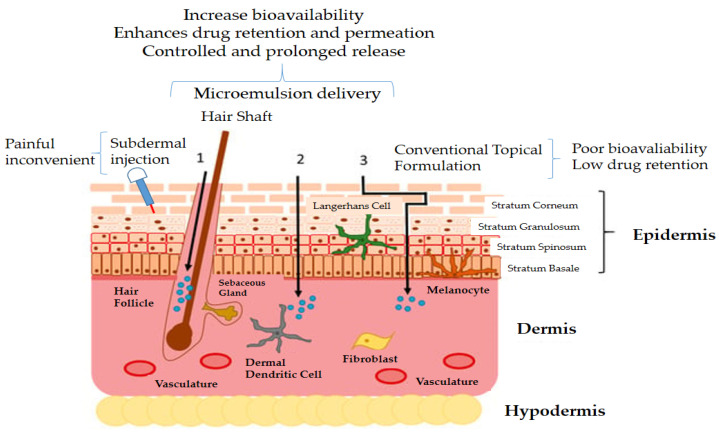
Representation of different routes of penetration of microemulsions through the skin: (1) by the appendageal route, (2) by the intracellular route, or (3) by the intercellular route. The appendageal route involves particles entering sweat glands, hair follicles, or skin furrows for either penetration into the dermis or retention for increased release of the drug. The intracellular route comprises a direct path via cell membranes containing multiple epidermal layers. The intercellular route comprises a more tortuous path between epidermal cells. The path taken by particles depends on dimensions, charge, shape, and material. Reproduced with permission from [135]. Copyright 2023, Multidisciplinary Digital Publishing Institute.

**Table 1 nanomaterials-13-01688-t001:** Some examples of microemulsions for skin delivery.

API	ME Type	Aqueous Phase	Surfactant/Co-Surfactant (S and Co-S)	Oil Phase	Application	Ref.
Sulfamerazine Indomethacin	O/W	Water Or PBS	**S:** Phosphatidylcholine; sodium oleate; Eumulgin^®^ HRE40; **Co-S:** N/A	Soybean oil	Antimicrobial	[137]
Curcumin	W/O	Water	**S:** Labrasol^®^, glyceryl oleate **Co-S:** propylene carbonate	Isopropyl palmitate	Anti-inflammatory	[138]
Pseudolaric acid B	O/W	Water	**S:** Cremphor EL^®^ **Co-S:** Transcutol P^®^	Isopropyl Myristate	Fungicidal	[139]
Pentoxifylline	W/O	Water	**S:** Tween 80; Brij 52 **Co-S:** N/A	Caprylic/Capric Triglycerides	Inflammatory dermatological diseases	[140]
Clobetasol	O/W	Water	**S:** Cremophor EL **Co-S:** isopropyl alcohol	Isopropyl myristate	Vitiligo	[141]
Betahistine hydrochloride	W/O	Water	**S:** Capryol 90^®^ **Co-S:** Transcutol^®^	Ethyl oleate	Regulation of feeding behavior and weight control	[142]
Ropivacaine	O/W	Water	**S:** Labrasol^®^ **Co-S:** Ethanol	Capryol^®^ 90	Anesthesia	[143]
Penciclovir	O/W	Water	**S:** Cremorphor EL **Co-S:** Ethanol	Oleic acid	Antiviral	[144]
Zidovudine	O/W	Water	**S:** Labrosol **Co-S:** Oleic Plurol	Isopropyl myristate	Antiretroviral therapy	[145]
Curcumin	O/W	Water	**S:** Chitosan **Co-S:** Ethanol	Oleic acid	Rheumatoid arthritis anti-inflammatory	[146]
Diacetyl boldine (DAB)	W/O	Water	**S:** Solutol^®^ HS 15; ethanol and Lecithin **Co-S:** Propylene glycol (co-surfactant)	Medium-chain triglyceride	Melanoma skin cancer treatment	[147]
Lidocaine	O/W	Water	**S:** Polyoxyl 15 hydroxystearate **Co-S:** Ethanol	Ethyl oleate	Anesthetic therapy	[148]
Oxcarbazepine	W/O	Water/buffer solution	**S:** Tween 80; Labrasol **Co-S:** PEG 400; Transcutol^®^ P	Oleic acid and cineole	Antiepileptic drug for epilepsy	[149]
Acemetacin	O/W	Water	**S:** Transcutol HP; Labrafil M1944 CS **Co-S:** Ethanol	Isopropyl myristate	analgesic and anti-inflammatory	[150]
Imiquimod	O/W	Water	**S:** Phospholipids (PL) **Co-S:** Ethanol	Oleic acid	psoriasis anti-inflammatory	[151]
Etofenamate	O/W	Water	**S:** Cremophor EL or Span 80 with Tween 20 **Co-S:** Transcutol HP; Ethanol	Oleic acid	Osteoarthritis treatment	[152]
Diclofenac	W/O	Water	**S:** Labrasol^®^ **Co-S:** Labrafil^®^	Limonene	Non-steroidal anti-inflammatory drugs	[153]
Finasteride	W/O	Water	**S:** Poloxamer 124 **Co-S:** Transcutol P	Oleic acid	Androgenetic alopecia treatment	[154]
Finasteride-cinnamon	W/O	Reverse osmosis water	**S:** Tween^®^ 20 **Co-S:** Propylene glycol	Cinnamon oil	Androgenetic alopecia treatment	[155]
Gallic acid	W/O	Water	**S:** Labrasol^®^/HCO-40^®^ or Tween 80/Span 80 **Co-S:** Transcutol^®^ or Ethanol	Isopropyl myristate	Antioxidant activity	[156]
Dencichine	O/W	Water/[HOEIM]Cl	**S:** Tween 80/[BMIM]C_12_SO_3_ **Co-S:** Propylene glycol	Isopropyl myristate	Hemostatic activity	[157]
Curcumin	O/W	Water	**S:** Cremophor^®^ RH 40 **Co-S:** Transcutol P	Oleic acid/limonene	Anticancer/antioxidant…	[158]
Celecoxib	W/O	Water	**S:** Phosphatidylcholine; decylglucoside **Co-S:** Ethanol; propylene glycol	Monocaprylin	Breast cancer treatment	[159]
Nifedipine	O/W	PBS	**S:** PPG-5-Ceteth-20 **Co-S:** N/A	Oleic acid	Anti-hypertensive Activity	[160]
Celecoxib	W/O	Water	**S:** PEG-6 Caprylic/Capric Glycerides **Co-S:** PEG-35 castor oil or PEG-7 glyceryl cocoate	Isopropyl myristate	Anti-inflammatory drugs	[161]
Coenzyme Q10	O/W	Water	**S:** Cremophor EL^®^ **Co-S:** Transcutol^®^ HP	Isopropyl myristate	Antioxidant activity	[162]

**API:** active pharmaceutical ingredient; **ME:** microemulsion.

**Table 2 nanomaterials-13-01688-t002:** Advantages and limitations of microemulsions for drug delivery.

Advantages of Microemulsions	Limitations of Microemulsions
- Enhanced drug solubilization [24]; - Improved drug stability [93]; - Increased drug permeability [24]; - Sustained drug release [174]; - Versatile drug delivery [21]; - Biocompatibility [26] - Long shelf life [82] - Ease of formulation [93]; - Targeted drug delivery [19]; - Reduced toxicity [131]; - Versatility (oral, parenteral, and topical) [11]; - Stable thermodynamically after preparation [25,75]	- Toxicity of high surfactant and co-surfactant concentrations [57]; - Limited drug loading capacity [175]; - Risk of leakage, which can lead to drug loss and reduced efficacy [176]; - Regulatory issues (safety and efficacy testing before approval for human use) [24] - Sensitivity to environmental factors during the preparation steps (Temperature, pH, and salt concentration) [133]

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
