# Peer review of "Exploring the Versatility of Microemulsions in Cutaneous Drug Delivery: Opportunities and Challenges"

_nanomaterials, 2023, doi:10.3390/nano13101688_

Round 1

Reviewer 1 Report

This review provides an overview of the composition, preparation methods, characterization techniques, and application scope of various microemulsions. In addition, the advantages and limitations of microemulsions as drug delivery systems are discussed, as well as the role of microemulsions in improving skin drug delivery is analyzed. This is a very meaningful work, however, the following issues should be addressed before the paper is considered suitable for publication in Nanomaterials.

1.      There are numerous formatting errors in the references, please correct them promptly.

2.      All pictures in the manuscript are not clear, and the text size and font on the picture are inconsistent, please adjust the format.

3.      Table 2 is not only problematic in formatting but also confusing in content. Please revise the content of the form in time to make it more beautiful and organized.

4.      There are inconsistencies in the advantages and limitations of microemulsions mentioned in Table 3. For example, the thermal stability of microemulsions is mentioned earlier, but later it is stated that microemulsions are sensitive to temperature.

5.      The title of this manuscript is “Exploring the versatility of microemulsions in drug delivery: opportunities and challenges”, but there is very little content in the text that is relevant to the topic. Please add relevant content.

6.      The drug delivery function of microemulsions is mentioned extensively in the text. To support this statement, please cite the papers: Chem. Soc. Rev. 2017, 46, 7021; Chem. Soc. Rev. 2021, 50, 2839; Nano Lett. 2022, 22, 7588; Sci. China: Chem. 2023, 66, 613.

Extensive editing of English language required

Author Response

Dear colleague,

First, I appreciate your valuable feedback and insights, as they have greatly contributed to enhancing the quality of this review. Your input has been instrumental in refining and improving the content. Thank you for your comments, which have been invaluable in making this review more comprehensive and informative.
Please find in the attached file the answers to your questions and remarks.
kind regards,

Zouhair AIT TOUCHENTE

Reviewer 2 Report

This review article reported microemulsions with detailed formulation methods and applications in drug delivery field. The article discussed the selection of basic components and active pharmaceutical ingredients (API). Then, characterization techniques were introduced. The advantages of microemulsions for skin drug delivery were clearly presented. And challenges and opportunities were comprehensively summarized. Table 2 and table 3 are helpful, informative and well organized. As a result, I would like to suggest publish in Nanomaterials after minor revisions noted.

1.      The introduction section is too simple. Since the authors want to focus on skin drug delivery, more information specifically about skin drug or skin diseases should be introduced. For example, the current skin drug delivery methods, and why the skin drug delivery is important.

2.      It is suggested to have “cutaneous” in the title showing the focus of this review clearly.

3.      Toxicity or biocompatibility is an important factor for drug delivery system. Are there any characterization methods about toxicity or biocompatibility of microemulsions can be introduction in characterization section?

Author Response

(The authors gave the same response as above.)
